# A Novel Analog Front End with Voltage-Dependent Input Impedance and Bandpass Amplification for Capacitive Biopotential Measurements

**DOI:** 10.3390/s20092476

**Published:** 2020-04-27

**Authors:** Hajime Nakamura, Yuichiro Sakajiri, Hiroshi Ishigami, Akinori Ueno

**Affiliations:** 1Master’s Program in Electrical and Electronic Engineering, Graduate School of Engineering, Tokyo Denki University, Tokyo 120-8551, Japan; uenolab_nakamura@uenolab.jp (H.N.); 18KMJ15@ms.dendai.ac.jp (Y.S.); 19KMJ03@ms.dendai.ac.jp (H.I.); 2Department of Electrical and Electronic Engineering, Tokyo Denki University, Tokyo 120-8551, Japan

**Keywords:** capacitive biopotential measurement, non-contact measurement, electrocardiogram, electromyogram, analog front end, voltage-dependent resistance

## Abstract

This paper proposes a novel analogue front end (AFE) that has three features: voltage-dependent input impedance, bandpass amplification, and stray capacitance reduction. With a view to applying the AFE to capacitive biopotential measurements (CBMs), the three features were investigated separately in a schematic and mathematical manner. Capacitive electrocardiogram (cECG) or capacitive electromyogram (cEMG) measurements using the AFE were performed in low-humidity conditions (below 35% relative humidity) for a total of seven human subjects. Performance evaluation of the AFE revealed the following: (1) the proposed AFE in cECG measurement with 1.70-mm thick clothing reduced the baseline recovery time and root mean square voltage of respiratory interference in subjects with healthy-weight body mass index (BMI), and increased R-wave amplitude for overweight-BMI subjects; and (2) the proposed AFE in cEMG measurement of biceps brachii muscle yielded stable electromyographic waveforms without the marked DC component for all subjects and a significant (*p* < 0.01) increase in the signal-to-noise ratio. These results indicate that the proposed AFE can provide a feasible balance between sensitivity and stability in CBMs, and it could be a versatile replacement for the conventional voltage follower used in CBMs.

## 1. Introduction

As population aging is accelerating globally and high medical costs are becoming increasingly problematic, the concept of smart healthcare is gaining attention as an effective solution [1,2]. Smart healthcare is a health service system that uses technology such as wearable devices, the Internet of Things (IoT), and mobile internet to dynamically access information, connect people, materials and institutions related to healthcare, and then actively manage and respond to medical ecosystem needs in an intelligent manner [3]. Smart healthcare has the potential to provide comprehensive patient care in diverse environments including hospitals and home treatment [4]. Thus, smart healthcare is expected to reduce medical costs, ameliorate the quality of life and enrich the user experience [5].

There is growing interest in various types of wearable devices that detect physiological information such as the electrocardiogram (ECG) and electromyogram (EMG) [6,7]. Previously reported wearables devices include patches [8], wristbands [9,10,11], earrings [12], eyeglasses [13], fabric [14] and garments [15,16,17]. In clinical practice, the Holter monitor is well-known among existing and commercially available wearable devices; it is able to record ECG continuously for 24 to 48 h. The Holter monitor is a useful device to diagnose cardiovascular diseases, which is the main cause of death around the world. However, as is the case with existing biopotential measurement devices, the Holter monitor requires direct contact of adhesive electrodes with the skin. Attachment for a long time or frequent detachment of the electrodes can cause skin allergy or inflammation as the measurement period gets longer. Thus, they are not suitable for monitoring over a prolonged period. In this context, capacitive biopotential measurement (CBM) has been studied because of its ability to detect biopotential without the direct contact of electrodes with the skin.

CBM is a measurement method that exploits capacitive coupling, which is comprised of skin, insulator (or cloth), and electrodes. Some researchers have already successfully used CBM in detecting ECG [18,19], EMG [20,21,22] and electroencephalograms (EEG) [23,24,25] without direct skin contact. The application of CBM is not limited to wearables, but also extends to furniture or fixtures such as beds [26,27], chairs [28,29], toilet seats [30], bathtubs [31], car seats [32,33], airplane seats [34], and operating tables [35]. However, compared to conventional biopotential measurement, which is based on resistive coupling, CBM requires an analog front end (AFE) with ever-higher input impedance in the measuring system so as not to cause voltage loss in the capacitive coupling of each measuring site. Meanwhile, as the time constant of the measuring system is a product of the input impedance and coupling capacitance of the measuring site, the more the input impedance of AFE increases, the longer the time constant becomes. Consequently, ever-higher input impedance, which is necessary for sensitive CBM, prolongs the baseline recovery (BR) time. BR time is a period when measurement is not possible due to out-of-range voltage input, which is caused mostly by triboelectricity [36] and/or static electricity. Out-of-range input commonly develops in CBM due to the body movements of the user but also the movements of others around the user, particularly under low-humidity conditions. The involvement of chemical textiles and thick clothing (or insulator) as an interfacing material between the skin and electrode also causes the out-of-range input. Therefore, finding a compromise between sensitivity (i.e., ever-higher input impedance) and stability (i.e., lower input impedance resulting in a shorter time constant) is an important challenge to be addressed in AFE design to reinforce the practicality and versatility of CBM, as noted in the literature [37].

Another issue related to the ever-higher input impedance of AFE is an increased susceptibility to stray interference such as power-line noise and its harmonics. In many cases, the housing and lead wires around AFE adopt a driven shield for interference suppression. A double shield, which is a combination of the driven shield and ground shield, was applied to multi-layered fabric electrodes and was found to mitigate motion-induced artifacts as well as the stray interference [38]. Furthermore, active ground is well known to be highly effective in reducing power-line noise [28]. Nevertheless, technical demand for noise suppression is still a matter of concern in CBM. This is because CBM is expected to be used in much noisier settings. Based on these considerations, we proposed a novel AFE that can cope with the need to find a balance between sensitivity and stability in CBM, which can also contribute to noise reduction. The proposed AFE was validated by measuring capacitive ECG (cECG) and capacitive EMG (cEMG) in low-humidity conditions. The signals of the cECG and cEMG that were obtained using the proposed AFE were compared with those obtained using a AFE that had been used in previous studies [27,39,40,41,42].

## 2. Previous and Proposed Analog Front End

A bootstrapped voltage follower with capacitance feedback (BVF*_cf_*), which is shown in Figure 1a, was employed as an AFE in CBM in several previous studies [27,40,41,42]. This BVF*_cf_* was first proposed by Thakor et al. in 1980 for conventional ECG measurements [39]. Its complex input impedance ${\dot{Z}}_{in\_BVF}$ is expressed in Equation (1), which assumes that the characteristics of the operational amplifier (OP Amp) used in the circuit are ideal [43].
(1)$${\dot{Z}}_{in\_BVF}\left(f\right)=\left(R_{a1}+R_{a2}\right)+j2\pi fC_{a3}R_{a1}R_{a2}$$
where *R_a_*_1_, *R_a_*_2_ are resistance, and *C_a_*_3_ is the capacitance for feedback in Figure 1a, and *f* is frequency. According to Equation (1), the magnitude of ${\dot{Z}}_{in\_BVF}$ varies with the frequency component in the input signal (i.e., it is frequency-dependent) and can be approximated into two states using corner frequency $f_{ca}=\left(R_{a1}+R_{a2}\right)/\left(2\pi C_{a3}R_{a1}R_{a2}\right)$, as shown in Equation (2):(2)$$\left|{\dot{Z}}_{in\_BVF}\left(f\right)\right|=\sqrt{{\left(R_{a1}+R_{a2}\right)}^{2}+{\left(2\pi fC_{a3}R_{a1}R_{a2}\right)}^{2}}≅\{\begin{array}{rr} R_{a1}+R_{a2}, & f\ll f_{ca}\\ \left(2\pi C_{a3}R_{a1}R_{a2}\right)f, & f\gg f_{ca} \end{array}$$

For an input frequency much lower than $f_{ca}$, (e.g., the direct-current (DC) component), $\left|{\dot{Z}}_{in\_BVF}\right|$ is simply (*R_a_*_1_ + *R_a_*_2_) and contributes to shortening the BR time as reported in [41]. With a sufficiently higher frequency component than $f_{ca}$, $\left|{\dot{Z}}_{in\_BVF}\right|$ becomes ever-higher to enable CBM by the use of large *C_a_*_3_. However, positive proportionality to frequency in ${\dot{Z}}_{in\_BVF}$ is equivalent to inductive characteristics. Therefore, the use of BVF*_cf_* in CBM, combined with capacitance of the skin-electrode coupling can cause resonant ringing in the signal and lead to signal instability [43].

The proposed novel AFE shown in Figure 1b is a bootstrapped non-inverting amplifier with resistance feedback (BNA*_rf_*), which has three features: (F-1) voltage-dependent input impedance, (F-2) bandpass amplification, and (F-3) stray capacitance reduction. The first feature (F-1) is attained by a bootstrapping part composed of a resistor *R_b_*_1_, a varistor with voltage-dependent resistance *R_bv_* and a feedback resistor *R_b_*_3_ as shown in Figure 1b. Given the varistor has a non-linear resistance that varies with the applied voltage ***v****_vrst_* relative to threshold voltages ±*V_thr_* as follows:(3)$$R_{bv}\approx \{\begin{array}{cc} R_{h}, & if-V_{thr}<v_{vrst}<+V_{thr}\\ 0, & else \end{array}$$

Then the input impedance of BNA*_rf_*, $Z_{in\_BNA}$, also becomes non-linear and varies with the input voltage *v_i_* (i.e., it is voltage-dependent). By applying star-delta transformation to the bootstrapping part in Figure 1b, combining virtual short concept of ideal OP Amp, and substituting Equation (3) into the derived input impedance, we can obtain the following:(4)$$Z_{in\_BNA}=R_{b1}+R_{bv}+\frac{R_{b1}R_{bv}}{R_{b3}}\approx \{\begin{array}{cc} \frac{R_{b1}}{R_{b3}}R_{h}, & if-V_{thr}<\frac{R_{h}}{R_{h}+R_{b3}}v_{i}<+V_{thr}\\ R_{b1}, & else \end{array}$$

According to Equation (4), when *v_i_* is within a range between $\pm V_{thr}\left(1+R_{b3}/R_{h}\right)$, $Z_{in\_BNA}$ can be ever-higher enough for sensitive CBM by choosing a low resistance for *R_b_*_3_, a medium (or high) resistance for *R_b_*_1_, and a high resistance for *R_h_.* Since normal *v_i_* (i.e., the biopotential at each measuring site) is in the order of millivolts at most, *v_i_* falls within the range between $\pm V_{thr}\left(1+R_{b3}/R_{h}\right)$ by choosing a varistor with *V_thr_* in the order of volts. On the other hand, for *v_i_* outside the range, $Z_{in\_BNA}$ becomes simply *R_b_*_1_ (i.e., medium) and contributes to shortening the BR time (i.e., stability in CBM). So, that means that the voltage-dependent input impedance of the proposed BNA*_rf_* in Equation (4) can achieve a feasible balance between sensitivity and stability in the CBM. Furthermore, when the range between $\pm V_{thr}\left(1+R_{b3}/R_{h}\right)$ is narrower than that between power-supply voltages to the OP Amp, ±*V_CC_*, the bootstrapping part that includes the varistor also works as pre-over-voltage protection for the subsequent OP Amp, leading to increased durability of the proposed AFE.

The second feature (F-2) of the proposed BNA*_rf_* (i.e., bandpass amplification) is attained, as shown in Figure 1b by the combination of a parallel connection of *R_f_* and *C_f_*, and a series connection of *R_s_* and *C_s_*. With the use of complex impedances of these connections, ${\dot{Z}}_{f}$ and ${\dot{Z}}_{s}$, complex voltage gain ${\dot{A}}_{v}$ of the non-inverting amplifier can be expressed, as shown in Equation (5):(5)$${\dot{A}}_{v}\left(f\right)=1+\frac{{\dot{Z}}_{f}}{{\dot{Z}}_{s}}=1+\frac{R_{f}}{R_{s}}\cdot \frac{1}{\left(1+j2\pi fC_{f}R_{f}\right)}\cdot \frac{j2\pi fC_{s}R_{s}}{\left(j2\pi fC_{s}R_{s}+1\right)}$$

For Equation (5), we can define corner frequencies of $f_{cf}=1/\left(2\pi C_{f}R_{f}\right)$ and $f_{cs}=1/\left(2\pi C_{s}R_{s}\right)$, for ${\dot{Z}}_{f}$ and ${\dot{Z}}_{s}$, respectively. When we set the frequency band $\left(f_{cf}-f_{cs}\right)$ wide enough, ${\dot{A}}_{v}$ is approximated into two transition frequency regions and an in-between passband region, as follows:
(6)$$\dot{A_{v}}\left(f\right)\approx \{\begin{array}{rr} 1+\frac{R_{f}}{R_{s}}\cdot \frac{j\left(f/f_{cs}\right)}{j\left(f/f_{cs}\right)+1}, & f<f_{cs}\ll f_{cf}\\ 1+\frac{R_{f}}{R_{s}}, & f_{cs}\ll f\ll f_{cf}\\ 1+\frac{R_{f}}{R_{s}}\cdot \frac{1}{1+j\left(\frac{f}{f_{cf}}\right)}, & f_{cs}\ll f_{cf}<f \end{array}$$

For a frequency *f* between $f_{cs}$ and $f_{cf}$ in Equation (6), the amplification factor of BNA*_rf_* is a constant $\left(1+R_{f}/R_{s}\right)$. Thus, we can consider BNA*_rf_* as a non-inverting amplifier for the frequency band of concern. Meanwhile, for the lower or higher frequency region of the relevant band, the amplitude of BNA*_rf_* gradually changes from $\left(1+R_{f}/R_{s}\right)$ to 1 as frequency *f* ($<f_{cs}$) decreases or frequency *f* ($>f_{cf}$) increases, respectively. Therefore, BNA*_rf_* behaves as a bandpass amplifier. In general, frequency confinement in amplification increases a percentage of the desired signal with the target frequencies in the output. Since an AFE with ever-higher input impedance is susceptible to surrounding noise, this feature (i.e., bandpass amplification) of BNA*_rf_* contributes to improvement in the signal-to-noise ratio (SNR) in CBM signals.

The third feature (F-3) of the proposed BNA*_rf_*, (i.e., stray capacitance reduction*)* is only attained by inserting a capacitor *C_n_* as indicated in Figure 1b [44]. While stray capacitance *C_st_* in Figure 1b inevitably incurs leak current ${\dot{I}}_{lk}$ depending on the potential difference between the non-inverting input node (${\dot{V}}_{i}$) and ground (0) of the OP Amp, the inserted *C_n_* yields compensating current ${\dot{I}}_{cp}$ depending on the potential difference between the output node (${\dot{V}}_{o}$) and the non-inverting input node (${\dot{V}}_{i}$). As each current is obtained from the potential difference divided by impedance of the capacitance and the gain of the amplifier is given by Equation (6), remaining leak current ${\dot{I}}_{lk}^{’}$ for the concerned frequencies *f* (i.e., $f_{cs}\ll f\ll f_{cf}$) is expressed as follows:(7)$${\dot{I}}_{lk}^{’}={\dot{I}}_{lk}-{\dot{I}}_{cp}=j2\pi fC_{st}\left({\dot{V}}_{i}-0\right)-j2\pi fC_{n}\left({\dot{V}}_{o}-{\dot{V}}_{i}\right)=j2\pi f\left(C_{st}-\frac{R_{f}}{R_{s}}C_{n}\right){\dot{V}}_{i}$$

According to Equation (7), we can consider the inserted *C_n_* to be negative capacitance $-\left(R_{f}/R_{s}\right)C_{n}$, reducing stray capacitance *C_st_*. Since ${\dot{I}}_{cp}$ corresponds to positive feedback current, it should be smaller than ${\dot{I}}_{lk}$ (i.e., ${\dot{I}}_{lk}^{’}>0$) for the prevention of oscillation. Therefore, *C_n_* has to be set to satisfy Equation (8):(8)$$C_{n}<\frac{R_{s}}{R_{f}}C_{st}$$

## 3. Experimental and Analytical Methods

All experimental procedures were approved by the Human Life Ethics Committee of Tokyo Denki University. All subjects provided informed consent prior to participation in our experiments.

### 3.1. Implementation of Previous and Proposed AFEs for Evaluation

To evaluate the proposed AFE (BNA*_rf_*), cECG and cEMG were measured using BNA*_rf_* and using a previous AFE (BVF*_cf_*) as a reference. Both AFEs were prototyped using off-the-shelf components. The same type of OP Amp with high input impedance (Texas Instruments, OPA129, 10TΩ//1pF according to the specification sheet) was employed to prevent degrading the input impedance synthesized in parallel with the bootstrapping part. The values of the elements used for circuit implementation and their specifications are shown in Table 1. Corner frequencies for bandpass in BNA*_rf_* (i.e., $f_{cs}$ and $f_{cf}$) were separately designed and implemented for cECG and cEMG, as can be seen in the frequency-gain characteristics shown in Figure 2. We confirmed that the measured characteristics in Figure 2 agreed with the theoretical characteristics described in Equation (6), and that $f_{cs}$ and $f_{cf}$ can be designed independently in conjunction with the gain for the concerned frequency *f* ($f_{cs}\ll f\ll f_{cf}$).

### 3.2. Measurements of cECG through Thick Clothing in a Low-Humidity Environment

In order to verify compatibility between the sensitivity and stability of the proposed BNA*_rf_* in CBM, cECG signal was measured using BNA*_rf_* under some challenging conditions for the measurement, where 1.70-mm thick clothing was placed between the skin and electrode, and the relative humidity was set to lower than 35%. For the purpose of comparison, another cECG signal was measured using BVF*_cf_* under the same conditions in a sequential manner. Four subjects with different body mass indexes (BMIs), falling within the range of healthy-weight (18.5–25) or overweight (25–30) BMI, participated in this experiment. Subject information and the measurement conditions are presented in Table 2. Since the BMI is defined as the body mass divided by the square of the body height and is universally expressed in units of kg/m^2^, the BMI of the subject under the force of gravity correlates roughly with the pressure of the capacitive coupling between the subject’s dorsal surface in a supine position and the electrode placed under the subject via the clothing. We selected these subjects to examine the effect of the coupling pressure on the quality of the obtained cECG signal. In the experiment, the subjects were requested to sit on a bed for 10 s, change their posture to a supine position on cue, and remain lying supine on the bed for 12 min with a constant breathing rate at 0.25 Hz in tune with the sound from a metronome.

As shown in Figure 3, a flexible sheet electrode (FSE), which was modified from previous studies [27,45], was used for the cECG measurements. The FSE is constructed from thin, soft conductive fabric (CSTK, Kitagawa Industries) and insulating textiles (100% polyester), and it has a total thickness of 0.45 mm. The FSE was placed under bed sheets made from 0.31 mm cotton, and underneath the back of the lying subject. All subjects wore a commercially available sweatshirt (96% cotton, 4% polyurethane) with a thickness of 0.84 mm, and an undershirt (66% cotton, 34% polyester) with a thickness of 0.55 mm. Therefore, the total thickness of the interface clothing between the skin and electrode was set to 1.70 mm.

The cECG signal was measured using the system shown in Figure 4. The reference Lead II ECG and respiratory movement (RM) signals were measured simultaneously with a commercially available telemetry system (BN-RSPEC, BIOPAC Systems). The frequency band of the circuits after the AFE was set to 0.5–100 Hz. The gain in the circuits was set to 40 dB in the measurement using the proposed BNA*_rf_*, but to 60 dB in the measurement using BVF*_cf_*, so that the total gain of the measurement system including AFE became 60 dB in both measurements.

### 3.3. Analysis of cECG Signals for Evaluation

Measured voltage of cECG signals was converted to input-referred voltage as a preprocessing common to BVF*_cf_* and BNA*_rf_*. As the first step to determine BR time for each cECG measurement, the baseline component was extracted from each cECG signal by applying a 0.15 Hz low-pass filter (LPF). Next, the steady-state baseline level (i.e., bias voltage) was computed as the mean of the baseline component from 60 s to 80 s after the onset of lying down on the bed. Finally, BR time was determined as the time required for the baseline component to return to a range of ±0.5 mV of the steady-state baseline level and to stay within the range for more than 10 s. A shorter BR time reflects the higher stability of the AFE used in the measurement against out-of-range input.

In order to evaluate the insusceptibility of the used AFE to RM disturbance in cECG measurement, the RM component around 0.25 Hz was extracted from each cECG signal by applying a 0.15–0.5 Hz bandpass filter. For each time-series RM component, the root mean square voltage (RMSV_resp_) was calculated by the minute from the onset of lying down on the bed. A smaller RMSV_resp_ means the used AFE has higher insusceptibility to RM disturbance.

### 3.4. Measurements of cEMG in a Low-Humidity Environment

In order to compare the performance of the previous and proposed AFEs in terms of SNR, the cEMG signal was measured in a sequential manner using either AFE where the relative humidity was lower than 35%. Low-humidity conditions are known to easily cause low-SNR output in CBM. The cEMG measurement was also executed to confirm the versatility of the proposed AFE. Three male subjects participated to this experiment. Subject information and the measurement conditions are summarized in Table 3. Each subject wore a commercially available long-sleeve cotton shirt, which was 0.26 mm thick. A flexible pad electrode shown in Figure 5 was fixed on the right biceps brachii muscle over the shirt by using a bandage. An additional driven ground electrode (DGE) was fixed near the right elbow, also over the shirt. The pad electrode was constructed from the same materials as the sheet electrode for the cECG measurements. The configuration of the pad electrode was modified based on studies in the literature [21,42,45].

In this experiment, each subject was requested to remain sitting in a chair and hold a 3 kg dumbbell in his right hand for 110 s with his right upper arm straight down at his side. While holding the dumbbell, the subject was instructed to bend his right elbow about 90 degrees with his upper arm straight down at his side, maintain the position for 10 s, and straighten the elbow again. This instruction was provided three times at 30 s, 50 s and 70 s in the 110 s experiment. This experiment was repeated twice for each subject, one using the previous AFE and the other using the proposed AFE.

During the experiment, cEMG signal was measured using the system shown in Figure 4. The frequency band of the circuit after the AFE was set to 20–500 Hz by combining the DC suppression circuit and LPF, as shown in the figure. The gain of the circuit after the AFE was adjusted so that the total gain of the measuring system became 60 dB, irrespective of AFE type. The reference EMG (EMG_ref_) signal was simultaneously measured with a commercially available EMG amplifier (BA1104M, TEAC Instruments), telemetry system (TU-4, TEAC Instruments), and disposable electrodes (F-150S, Nihon Kohden). The disposable electrodes were directly attached on the skin near the right biceps brachii muscle so that they did not interfere with the pad electrode for cEMG measurement.

### 3.5. Analysis of cEMG Signals for Evaluation

The measured voltage of cEMG and EMG_ref_ signals was converted to input-referred voltage as a preprocessing similar to that described in 3.3. For cEMG signals, the 10 s period of “elbow bended” and the subsequent 10 s period of “elbow straightened” were considered as a pair, and two segments of 4000 samples, from 4 s to 8 s (${v_{i}}^{\left(S+N\right)}$) and from 14 s to 18 s (${v_{i}}^{\left(N\right)}$) were extracted from the one pair. A total of twelve segments were extracted from one subject; 6 (i.e., 3 pairs) from signals measured with the previous AFE, and the other 6 from signals measured with the proposed AFE. To compare the signal quality of cEMG of the two AFEs, digital Fourier transform (DFT) was applied to each segment. In addition, SNR was computed from each pair of the two segments as follows:(9)$$\mathrm{SNR}\left(v\right)=20\log_{10}\frac{\sqrt{\frac{1}{n}\sum_{i=1}^{n}{\left({v_{i}}^{\left(S+N\right)}-{v_{i}}^{\left(N\right)}-{V_{Bias}}^{\left(S\right)}\right)}^{2}}}{\sqrt{\frac{1}{n}\sum_{i=1}^{n}{\left({v_{i}}^{\left(N\right)}-{V_{Bias}}^{\left(N\right)}\right)}^{2}}}\;\left(\mathrm{dB}\right)$$
(10)$${V_{Bias}}^{\left(S\right)}=\frac{1}{n}\overset{n}{\underset{i=1}{\sum }}\left({v_{i}}^{\left(S+N\right)}-{v_{i}}^{\left(N\right)}\right)$$
(11)$${V_{Bias}}^{\left(N\right)}=\frac{1}{n}\overset{n}{\underset{i=1}{\sum }}{v_{i}}^{\left(N\right)}$$

We considered that the “elbow bended” segments contain both myoelectric signal and noise (*S+N*), while “elbow straightened” segments include noise only (*N*).

## 4. Results

### 4.1. cECG Measurements through Thick Clothing in a Low-Humidity Environment

Figure 6 shows the contrast in the transitions of cECG signals measured with either the previous or the proposed AFEs for two subjects #A and #C with different BMIs. In spite of a challenging environment for the measurement, both systems were able to detect visible cECG waveforms from all subjects by the end of the 12 min measurement period. For the subjects within a healthy-weight BMI range (i.e., #A, #D), however, the systems showed a big difference in the stability of the signal baseline, as can be seen in Figure 6a,b. The baseline of the cECG signal obtained with the previous AFE (Figure 6a) fluctuated, even at the end of the measurement, whereas that obtained with the proposed AFE (Figure 6b) fluctuated moderately. Consequently, T waves as well as R waves became visible toward the end of the measurement period, as can be seen in Figure 6b. The fluctuations of both seemed to be cyclic with a period of around 4 s, inferring an association with the 0.25-Hz breathing.

Meanwhile, for the subjects in the overweight BMI range (i.e., #B, #C), there was little difference in the baseline stability of the systems, as shown in Figure 6c,d. Using either the previous or proposed AFEs, the baseline of the cECG signal was stable from the beginning in each recording. In terms of the R wave amplitude, the system with the proposed AFE detected larger amplitude and was slight superior because of its higher input impedance.

### 4.2. Evaluation of cECG Signals

Figure 7 shows examples of the transient responses of the baseline to the onset of subject (#A) with healthy-weight BMI lying down. With the use of the proposed AFE, the BR time decreased from 42.4 s to 8.9 s. As in the case of the contrasting baseline responses in Figure 6, the BR times in Table 4 were significantly different for subjects #A and #D, who are within a healthy-weight BMI range. However, for subjects #B and #C who are in the overweight BMI range, the BR time showed little difference. For all subjects, the BR times with the proposed AFE were shorter than 10 s.

Figure 8 shows minute-by-minute transitions of RMSV_resp_ extracted from cECG signals measured with the previous or proposed AFE for all four subjects. Comparing the transitions between the subjects irrespective of AFE type, transitions in the subjects within the overweight BMI range (#B, #C) dropped rapidly after 1 min as in Figure 8b,c. On the other hand, for the subjects within the healthy-weight BMI range, transitions decreased gradually with time as can be seen in Figure 8a,d. These figures, show that compared with the previous AFE, the RMSV_resp_ obtained with the proposed AFE in every one minute was smaller in most times. Having grouped minute-by-minute twelve RMSV_resp_ data for each subject together to compare the two AFEs, the RMSV_resp_ with the proposed AFE was significantly smaller (*p* < 0.01) than that with the previous AFE for both subjects #A and #D (data not shown). These results were consistent with the results in Figure 6 and Figure 7 and Table 4.

### 4.3. Measurements of cEMG in a Low-Humidity Environment

Figure 9 shows example of EMG_ref_ and cEMG recordings measured with the previous and the proposed AFE under low-humidity conditions for subject #E. For both of the AFEs, electromyographic firings during the “elbow bended” period in the cEMG signals were synchronized with those in the simultaneously measured EMG_ref_ signals. However, at times around elbow bending (t = 0 s) and elbow straightening (t = 10 s), there was a specific difference between the two AFEs in terms of baseline stability. When the previous AFE was used, the baseline of cEMG fluctuated widely around these timings, whereas the baseline settled with the use of the proposed AEF. This superiority in the baseline stability of the proposed AFE in the cEMG signal was observed in all subjects. In addition, with the use of the proposed AFE, the undesirable DC component in the signal was close to 0 in all subjects.

### 4.4. Evaluation of cEMG Signals

Figure 10 shows the DFT spectra of the analyzed segments in the cEMG signals depicted in Figure 9. When comparing the undesirable low-frequency components (< 20 Hz) of the two AFEs, the spectral amplitudes with the proposed AFE were obviously smaller than those of the previous AFE, for both the elbow-bended segments (blue lines) and the elbow-straightened segments (red lines). On the other hand, when comparing the target frequency components (20–500 Hz), the spectral amplitudes with the proposed AFE for the elbow-bended segment were higher than that with the previous AFE. As for the elbow-straightened segment, the harmonic amplitudes of power-line frequency (50 Hz in the east area of Japan) with the proposed AFE apparently decreased. These differences, which show the superiority of the proposed AFE, were consistent with the results shown in Figure 9, and were common in all subjects.

Figure 11 shows a comparison of the SNR of cEMG of the two AFEs. As shown in Figure 11a, the SNR with the proposed AFE clearly increased in each subject. Consequently, by grouping the SNR data for all subjects together, a comparison of the SNR between the AFEs indicated significance (*p* < 0.01) as shown in Figure 11b. These results are consistent with the results of the DFT spectra in Figure 10, which showed larger amplitudes in the frequency passband for the elbow-bended segment and smaller amplitudes for the elbow-straightened segment with the proposed AFE.

## 5. Discussion

The results depicted in Figure 6, Figure 7, Figure 8, Figure 9, Figure 10 and Figure 11, confirm the superiority of the proposed AFE, which can be attributed to its three features: (F-1) voltage-dependent input impedance, (F-2) bandpass amplification, (F-3) stray capacitance reduction. We discuss the results in terms of the three features in the following subsections.

### 5.1. Significance of Voltage-Dependent Input Impedance

A comparison of the transient response of the cECG baseline shown in Figure 7a,b indicates that the proposed AFE has an advantage in its quick discharge capability owing to the voltage-dependent input impedance. In low-humidity conditions and the use of clothing made of chemical fiber in the skin-electrode coupling in CBM, our body and clothing are easily be electrically charged. Given a coupling charge between the skin and electrode for the measurement is $Q_{C}$ at the beginning of CBM, the coupling with capacitance $C_{C}$ generates DC voltage $V_{C}=Q_{C}/C_{C}$. Therefore, any new formation of the coupling, for instance, by lying down on the in-bed electrode during cECG measurements, can incur a large-magnitude input voltage ($>\left|\pm V_{thr}\left(1+R_{b3}/R_{h}\right)\right|$) to the AFE of the measurement system. Similarly, an increase of $Q_{C}$, for instance, by friction caused by bending or straightening motions in cEMG measurements, can also produce a large-magnitude input. Therefore, because of the lower input impedance of the proposed AFE against the large-magnitude input $V_{C}$, the BR times were shorter for subjects #A and #D, as shown in Figure 7b and Table 4. For the same reason, moderate amplitude in the cEMG signal was also attained at the beginning of the bending motion, as shown in Figure 9b. In addition, in regard to the BR times for subject #B and #C, the DC voltage $V_{C}$ can be transformed to Equation (12) using the coupling distance $d_{C}$, coupling dielectric constant $\varepsilon_{C}$, and coupling area $A_{C}$ as follows:(12)$$V_{C}=\frac{Q_{C}}{C_{C}}=\frac{Q_{C}}{\varepsilon_{C}A_{C}}d_{C}$$

According to Equation (12), a small $d_{C}$ resulting from a large BMI (i.e., large coupling pressure) can result in small $V_{C}$ input to the AFE. In the case where the magnitude of $V_{C}$ is smaller than $\left|\pm V_{thr}\left(1+R_{b3}/R_{h}\right)\right|$, the input impedance of the proposed AFE is comparable to that of the previous AFE, leading to BR times in the same range in subjects #B and #C, as seen in Table 4.

When considering the fluctuation of $V_{C}$, we can derive Equation (13) by differentiating Equation (12) regarding $\varepsilon_{C}$ and $A_{C}$ as constants.
(13)$$\frac{dV_{C}\left(t\right)}{dt}=\frac{1}{\varepsilon_{C}A_{C}}\left(\frac{dQ_{C}\left(t\right)}{dt}d_{C}\left(t\right)+Q_{C}\left(t\right)\frac{dd_{C}\left(t\right)}{dt}\right)$$

While the first member in Equation (13) represents the discharging process of $Q_{C}$ associated with BR time, the second member corresponds to the fluctuation caused by the change in $d_{C}$ such as the regulated breathing motion executed during cECG measurements. We can see that the second member is proportional not only to change in $d_{C}$ but also to $Q_{C}$ at the time. Therefore, residue of $Q_{C}$ resulting from the discharge can affect RMSV_resp_, which reflects the amplitude of disturbance caused by respiratory motion in cECG signals. In Figure 8a,d, compared with Figure 8b,c, larger values of RMSV_resp_ were obtained during 2–12 min periods. Presumably, these were because the lower coupling pressure of the healthy-weight-BMI subjects, #A and #D, resulted in a longer discharge process and in larger residue of $Q_{C}\left(t\right)$ in each minute. Comparison of the RMSV_resp_ values in subject #A and #D for the two AFEs shows the superior capability of the proposed AFE for quick discharge at low coupling pressure.

The oscillatory fluctuations observed in the cEMG signal (shown in Figure 9a) at the beginning of elbow bending and elbow straightening have the potential for resonance phenomena. As noted in Section 2 and in the literature [43], series connection of capacitive coupling with $C_{C}$ and BVF_cf_ (i.e., the previous AFE) can cause resonant ringing at a frequency $f_{RS}$, as shown in Equation (14):(14)$$f_{RS}=\frac{1}{2\pi \sqrt{C_{C}C_{a3}R_{a1}R_{a2}}}$$

Given $C_{C}$ is 1.0 pF, assigning values in Table 1 to $C_{a3}$, $R_{a1}$ and $R_{a2}$ leads to $f_{RS}≅0.16$ Hz. Though this frequency does not correspond with the actual frequency observed at the beginnings of the motion, they can be a ringing response of the high-pass filter (DC suppression circuit) in the measurement system (in Figure 4) against large stepwise input caused by the resonance. The undesirably large amplitudes in the low frequency region (< 10 Hz) in both spectrums, shown in Figure 10a are also attributable to the resonance. The proposed AFE also has an advantage in terms of preventing resonance.

### 5.2. Usefullness of Bandpass Amplification in AFE

Comparison of the baseline in the cEMG signal in Figure 9a,b suggests another advantage of the proposed AFE in bandpass amplification. The baseline in Figure 9a is biased about −0.7 mV with the use of the previous AFE, whereas that in Figure 9 was vanishingly biased with the proposed AFE. This small bias is probably due to unequal gains in the proposed AFE for the DC component and for the components with the target frequencies (20–500 Hz). We can confirm that the gain for the DC component is 0 dB by observing Figure 2 and by assigning zero to *f* in Equation (6). On the other hand, the gain for the components with the frequencies of interest is 20 dB in the proposed AFE but it is 0 dB in the previous AFE. Consequently, even when both measurement systems have the same total gain (i.e., 60 dB) for the components with the frequency of concern, the relative magnitude of the DC component decreases in the system with the proposed AFE. Similar logic can be applied to the components with non-DC frequencies of no concern. Smaller amplitudes in lower frequency (< 20 Hz) in the spectrum shown in Figure 10b are consistent with this logic. Furthermore, generally speaking, limiting the frequency band reduces thermal noise. Therefore, the bandpass feature of the proposed AFE was considered to be an important factor in the higher SNR in cEMG signal, as shown in Figure 11.

### 5.3. Effect of Stray Capacitance Reduction

Comparison of the R-wave amplitudes in Figure 6c,d suggests the presence of difference in input impedance between the two AFEs. As noted in previous research [19], stray capacitance between the measuring electrode and circuit ground in CBM can cause attenuation of R-wave amplitude. Consequently, larger R-wave amplitudes in Figure 6d, imply less stray capacitance in the input impedance, are attributable to *C_n_* being introduced only in the proposed AFE. Similarly, larger amplitudes of cEMG in the spectrum for the elbow-bended segment in Figure 10b are also due to reduced stray capacitance by *C_n_* in the proposed AFE. Note that the introduction of *C_n_* in the previous AFE does not reduce stray capacitance because compensating current is not produced due to no potential difference between the output and input nodes of the voltage follower. Hence, a change in AFE type from a voltage follower to a non-inverting amplifier is also necessary for the introduction of *C_n_*.

## 6. Conclusions

This study proposed a novel AFE that has three features, which include voltage-dependent input impedance, bandpass amplification, and stray capacitance reduction. With a view to applying the AFE to CBMs, the three features were separately investigated in a schematic and mathematical manner. Measurements of cECG or cEMG using the AFE were performed under low-humidity conditions (below 35% relative humidity) for a total of seven human subjects. Performance evaluation of the AFE revealed the following: (1) The proposed AFE in cECG measurements with 1.70 mm thick clothing reduced the BR time and RMSV_resp_ for healthy-weight-BMI subjects, and increased the R-wave amplitude for overweight-BMI subjects. (2) The proposed AFE in cEMG measurements of biceps brachii muscle yielded stable electromyographic waveforms without a visibly biased baseline for all subjects and a significant (*p* < 0.01) increase in SNR. These results indicate that the proposed AFE can provide a feasible balance between sensitivity and stability in the CBM, and could be a versatile replacement for conventional voltage followers used in CBMs.

## Figures and Tables

**Figure 1 sensors-20-02476-f001:**
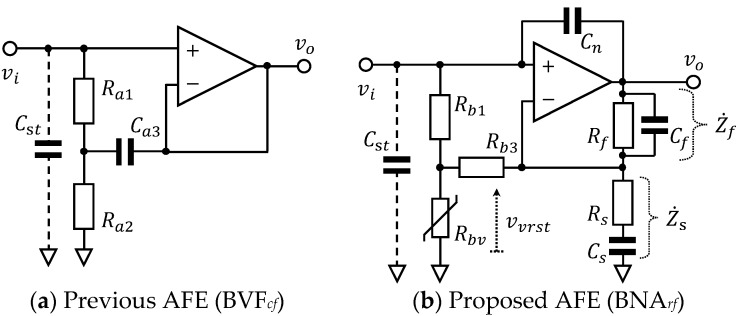
Configuration of analogue front ends (AFEs) involving bootstrapping techniques: (**a**) Previous AFE (BVF*_cf_*) bootstrapped by resistors *R_a_*_1_ and *R_a_*_2_, and a feedback capacitor *C_a_*_3_. This serves as a voltage follower; (**b**) Proposed AFE (BNA*_rf_*) bootstrapped by resistors *R_b_*_1_ and *R_b_*_3_, and a varistor with voltage-dependent resistance *R_bv_*. This serves as a non-inverting bandpass amplifier by the use of resistors *R*_s_ and *R_f_*, and capacitors *C_s_* and *C_f_*. A capacitor *C_n_* is incorporated to reduce stray capacitance *C_st_*.

**Figure 2 sensors-20-02476-f002:**
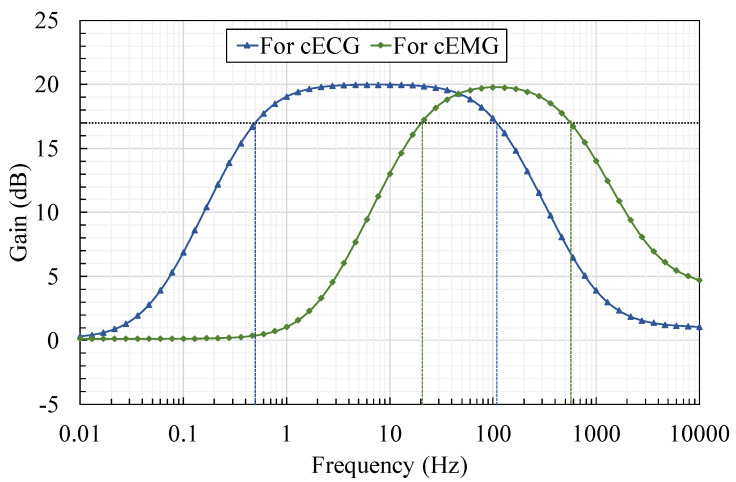
Frequency-gain characteristics of the proposed AFE with bandpass amplification tuned separately for cECG and cEMG measurements. Maximum gains were set to 20 dB for both measurements. Frequency passbands were set to 0.5–103 Hz for cECG and to 20–513 Hz for cEMG, respectively.

**Figure 3 sensors-20-02476-f003:**
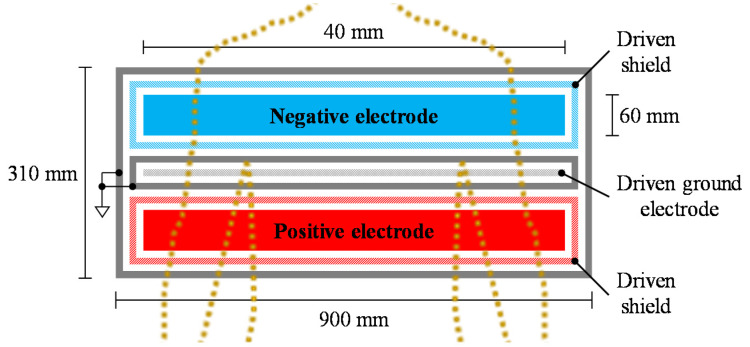
Top view configuration of flexible sheet electrode (FSE) for cECG measurement. The configuration was modified from [27,45].

**Figure 4 sensors-20-02476-f004:**
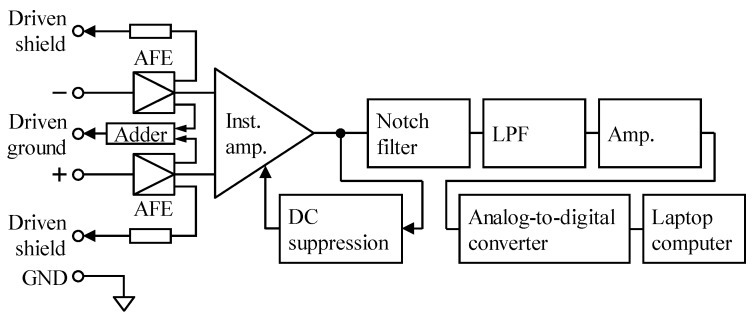
Schematic diagram of the measuring system for both cECG and cEMG. Total gain including AFE was set to 60 dB according to the type of AFE. The frequency bands achieved by a combination of DC suppression and low-pass filter (LPF) were set to 0.5–100 Hz for cECG measurement and to 20–500 Hz for cEMG measurement, respectively. The analog signal was digitized at 1000 Hz with 16-bit resolution between −10 and +10 V by the analog-to-digital converter.

**Figure 5 sensors-20-02476-f005:**
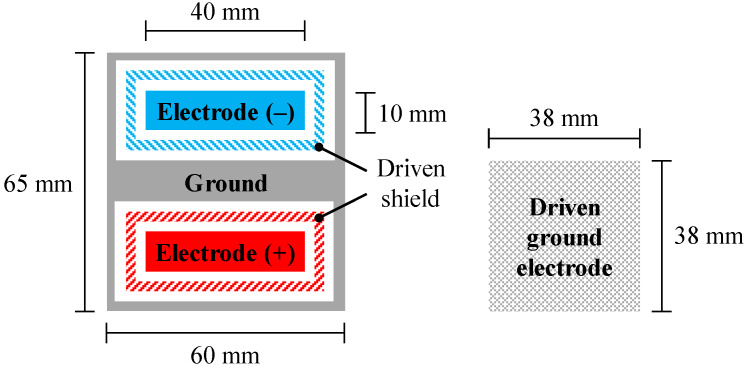
Top view configuration of flexible pad electrode (**left**) and driven ground electrode (DGE, **right**) employed for cEMG measurement. The pad electrode has a five-layered structure. DGE is used for the negative feedback of a voltage synthesized from two voltages that were independently obtained with negative and positive electrodes in the pad.

**Figure 6 sensors-20-02476-f006:**
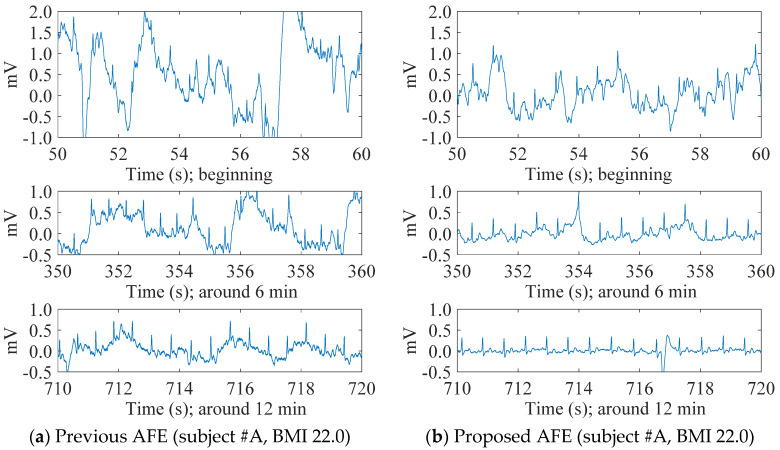
The transition of cECG recordings measured through thick clothing (1.70 mm) in low-humidity environments. The previous AFE was used for the recordings of (**a**,**c**), whereas the proposed AFE was used for the recordings of (**b**,**d**). Both recordings (**a**,**b**) were measured from a healthy-weight-BMI subject (22.0, #A), and recordings (**c**,**d**) were from an overweight-BMI subject (28.6, #C).

**Figure 7 sensors-20-02476-f007:**
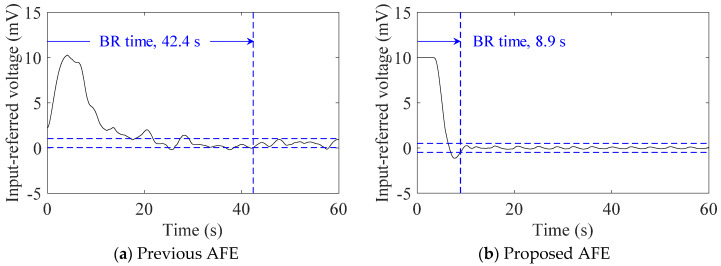
Examples of transient response of cECG baseline to the onset of supine position of subject #A. Original cECG signals were measured using the (**a**) previous AFE and (**b**) proposed AFE, and then filtered to extract their baselines, respectively. The baseline recovery (BR) time is also indicated in each response.

**Figure 8 sensors-20-02476-f008:**
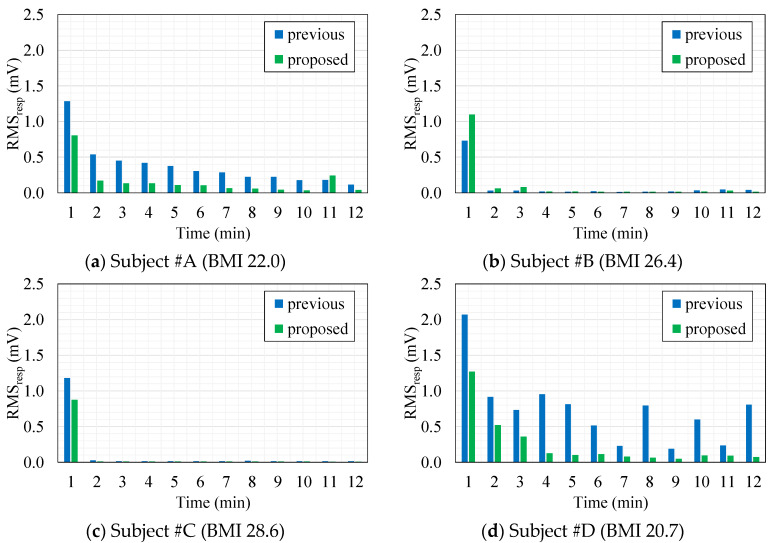
Minute-by-minute transitions of the root mean square voltage (RMSV_resp)_ extracted from cECG recordings measured with the previous AFE (blue) or the proposed AFE (green) in each subject. (**a**) Subject #A (BMI 22.0). (**b**) Subject #B (BMI 26.4). (**c**) Subject #C (BMI 28.6). (**d**) Subject #D (BMI 20.7).

**Figure 9 sensors-20-02476-f009:**
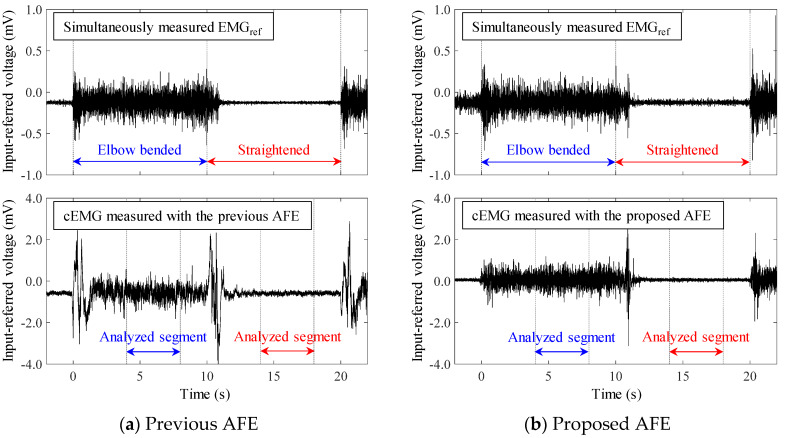
Examples of simultaneously measured reference EMG (EMG_ref_) and cEMG through cotton cloth (0.26 mm thick) under conditions of low-humidity (19%, 3.91 g/m^3^) from subject #E using (**a**) the previous AFE and (**b**) the proposed AFE.

**Figure 10 sensors-20-02476-f010:**
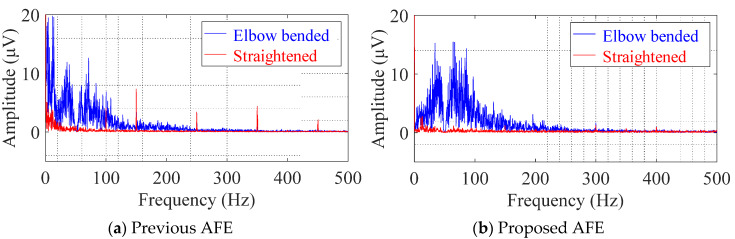
Digital Fourier transform (DFT) spectra for cEMG recordings in Figure 9 obtained from subject #E using (**a**) the previous AFE and (**b**) the proposed AFE. The blue line corresponds to the spectra for “Elbow bended” segments, and the red line corresponds to that for “Elbow straightened” segments.

**Figure 11 sensors-20-02476-f011:**
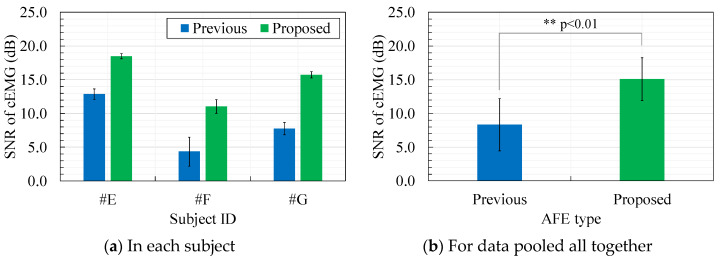
Comparison of the signal-to-noise ratio (SNR) of recorded cEMG between the previous and proposed AFEs: (**a**) Comparison for each subject, (**b**) Comparison for data gathered from all subjects.

**Table 1 sensors-20-02476-t001:** Specifications and values of the elements used in the implementation of the previous and proposed AFEs for capacitive electrocardiogram (cECG) and capacitive electromyogram (cEMG) measurements.

Item	Previous AFE (BVF*_cf_*)			Proposed AFE (BNA*_rf_*)		
Gain	Total 60 dB	AFE 0 dB	Subsequent 60 dB	Total 60 dB	AFE (*R_s_*, *R_f_*) 20 dB (30 k, 270 k Ω)	Subsequent 40 dB
Bootstrap	*R_a_*_1_ 100 MΩ	*R_a_*_2_ 100 MΩ	*C_a_*_3_ 100 µF	*R_b_*_1_ 100 MΩ	*R_bv_* 20 MΩ/0 Ω	*R_b_*_3_ 1.0 kΩ
Bandpass		NA		Measurand cECG cEMG	*f_cs_* (*C_s_* µF) 0.50 Hz (10.7) 20.0 Hz (2.67)	*f_cf_* (*C_f_* nF) 103 Hz (5.70) 513 Hz (1.15)
StrayCap.Reduction		NA			*C_n_* 1.0 pF	

NA: not applicable, Cap.: Capacitance.

**Table 2 sensors-20-02476-t002:** Subject information and experimental conditions in cECG measurement.

ID	Subject Information				Clothing	Experimental Conditions		
	Height (m)	Weight (kg)	BMI (kg/m^2^)	Age	Thickness (mm)	RH (%)	Temperature (°C)	VH (g/m^3^)
#A	1.65	60	22.0	22	1.70	22.0	24.0	4.79
#B	1.63	70	26.4	23	1.70	22.0	24.0	4.79
#C	1.61	74	28.6	23	1.70	32.6	25.0	7.52
#D	1.69	59	20.7	23	1.70	17.0	24.8	3.88

BMI: body mass index, RH: relative humidity, VH: volumetric humidity.

**Table 3 sensors-20-02476-t003:** Subject information and experimental conditions in cEMG measurement.

ID	Subject Information				Clothing	Experimental Conditions		
	Height (m)	Weight (kg)	BMI (kg/m^2^)	Age	Thickness (mm)	RH (%)	Temperature (°C)	VH (g/m^3^)
#E	1.71	56	19.2	24	0.26	19.0	23.0	3.91
#F	1.80	92	28.4	23	0.26	19.0	23.0	3.91
#G	1.70	73	25.3	23	0.26	33.0	23.2	6.87

**Table 4 sensors-20-02476-t004:** Comparison of BR time(s) between the previous and proposed AFEs. The BMI of each subject and the humidity during each measurement are also indicated.

Subject ID	BR Time (s)		BMI (kg/m^2^)	Humidity	
	Previous	Proposed		RH (%)	VH (g/m^3^)
**#A**	42.4	8.9	22.0	22.0	4.79
#B	4.2	5.1	26.4	22.0	4.79
#C	6.6	6.4	28.6	32.6	7.52
#D	69.2	8.0	20.7	17.0	3.88
Mean ± SD	30.6 ± 26.9	7.1 ± 1.5	24.4 ± 3.7	23.4 ± 5.7	5.25 ± 1.37

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
