# Peer review of "A Novel Analog Front End with Voltage-Dependent Input Impedance and Bandpass Amplification for Capacitive Biopotential Measurements"

_sensors, 2020, doi:10.3390/s20092476_

Round 1

Reviewer 1 Report

The proposed front end for bio signal measurements is interesting in particular the stray capacitance reduction is particularly useful during the electromyogram measure. I have only minor suggestions aimed at improve the quality of the work, in particular:

1)figure 6 the y range should be increased the first graph is not visible. What about to normalise the signal in order to have a clear comparison?

2) the manuscript requires minor grammatical check, please do it.

Author Response

The authors would like to thank you for your helpful and insightful reading of our manuscript that helped to improve its clarity. We have modified the manuscript and we detail our response to the reviewer’s comments in the attached PDF file.

Reviewer 2 Report

Presented paper shows a novel AFE with Voltage-Dependent Input Impedance and Bandpass Amplification for Capacitive Biopotential Measurements  (eECG, eEMG). Presented novel AFE were tested using 7 subjects (4+3) with healthy and overweight BMI index and compared with previous AFE presented in the literature.

The reviewer very well assesses the quality of the article. The subject matter fits very well with current trends. However, the reviewer asks the authors to respond to minor comments:

  1. It should be clearly explained why people were selected for the study in terms of BMI. How this parameter can affect the measurement results.
  2. it is worth placing (for example as a supplementary material) the exact diagram (values of components and models of integrated circuits) used during the tests. This will allow practical use of the designed AFE by other researchers.
  3. In line 87 there are some problems with References. Please correct.

Author Response

(The authors gave the same response as above.)

Reviewer 3 Report

In this paper, the authors present a new analog front end for capacitive biopotential measurements. The proposed circuit is a bootstrapped voltage follower with resistive feedback (AFErf). Authors compared such a configuration with the one of a bootstrapped voltage follower with a capacitive feedback (AFEcf) and demonstrated that the former one provides a voltage dependent input resistance, a bandpass amplification, and a reduction of the stray capacitance. After a theoretical presentation, the two configurations were exploited to acquire capacitive ECG and EMG from four and three adult volunteers respectively. Data confirmed superiority of the proposed AFErf

Major revision:

Clearly declare that the study has been approved by your Institutional Review Board before trials on volunteers

Minor comments:

Pag 2, line 87: correct references

Pag 6, line 188: FSE not defined (add a definition at line 186). Move the sentence “ The FSE is constructed from… 045mm tick” from line 188-190 to line 187 after “…cECG measurments”. The current position is misleading because the sentence reports the thickness of the FSE but the focus of this part of the work is the thickness of the interface clothing.

Pag 8, lines 253-260: It could be useful to anticipate the definition of vi S+N e vi N to make easier the comprehension of the following equations (9, 10, and 11). For this reason, I suggest rephrasing lines 254-256 and lines 261-265 as in the following:

254-256: For cEMG signals, 10-s period of “elbow bended” and subsequent 10-s period of “elbow straightened” were considered as a pair, and two segments of 4000 sample, from 4s to 8 s  (vi S+N) and from 14 s to 18 s (vi N) were extracted from the one pair.(…)

261-265: We considered “elbow 263 bended” segments contain both myoelectric signal and noise (S+N), while “elbow straightened” segments include noise only (N).

Author Response

(The authors gave the same response as above.)
